# HSP-UNet: An Accuracy and Efficient Segmentation Method for Carbon Traces of Surface Discharge in the Oil-Immersed Transformer

**DOI:** 10.3390/s24196498

**Published:** 2024-10-09

**Authors:** Hongxin Ji, Xinghua Liu, Peilin Han, Liqing Liu, Chun He

**Affiliations:** 1School of Electrical Engineering, China University of Mining and Technology, Xuzhou 221116, China; ts22230107p31@cumt.edu.cn; 2College of Mechanical and Electronic Engineering, Shandong Agricultural University, Tai’an 271018, China; lxh9357@163.com; 3State Grid Tianjin Electric Power Research Institute, Tianjin 300180, China; liulq328@126.com (L.L.); hechuntjsgcc@163.com (C.H.)

**Keywords:** surface discharge, discharge carbon trace, image segmentation, semantic segmentation, UNet, HPA, SCA

## Abstract

Restricted by a metal-enclosed structure, the internal defects of large transformers are difficult to visually detect. In this paper, a micro-robot is used to visually inspect the interior of a transformer. For the micro-robot to successfully detect the discharge level and insulation degradation trend in the transformer, it is essential to segment the carbon trace accurately and rapidly from the complex background. However, the complex edge features and significant size differences of carbon traces pose a serious challenge for accurate segmentation. To this end, we propose the Hadamard production-Spatial coordinate attention-PixelShuffle UNet (HSP-UNet), an innovative architecture specifically designed for carbon trace segmentation. To address the pixel over-concentration and weak contrast of carbon trace image, the Adaptive Histogram Equalization (AHE) algorithm is used for image enhancement. To realize the effective fusion of carbon trace features with different scales and reduce model complexity, the novel grouped Hadamard Product Attention (HPA) module is designed to replace the original convolution module of the UNet. Meanwhile, to improve the activation intensity and segmentation completeness of carbon traces, the Spatial Coordinate Attention (SCA) mechanism is designed to replace the original jump connection. Furthermore, the PixelShuffle up-sampling module is used to improve the parsing ability of complex boundaries. Compared with UNet, UNet++, UNeXt, MALUNet, and EGE-UNet, HSP-UNet outperformed all the state-of-the-art methods on both carbon trace datasets. For dendritic carbon traces, HSP-UNet improved the Mean Intersection over Union (MIoU), Pixel Accuracy (PA), and Class Pixel Accuracy (CPA) of the benchmark UNet by 2.13, 1.24, and 4.68 percentage points, respectively. For clustered carbon traces, HSP-UNet improved MIoU, PA, and CPA by 0.98, 0.65, and 0.83 percentage points, respectively. At the same time, the validation results showed that the HSP-UNet has a good model lightweighting advantage, with the number of parameters and GFLOPs of 0.061 M and 0.066, respectively. This study could contribute to the accurate segmentation of discharge carbon traces and the assessment of the insulation condition of the oil-immersed transformer.

## 1. Introduction

Large oil-immersed transformers play a critical role in the power system [1]. Their failure would affect the power supply of the entire system and even have serious social consequences. Preventive overhaul and maintenance of the transformers are essential for the stable operation of the power system [2,3,4]. Due to the metal-enclosed shell and the complex internal structure, internal defects of large transformers are difficult to detect. Commonly used methods, such as manually drilling into the transformer and lifting the shell, face the problems of low efficiency, poor accuracy, high risk and high cost. With the rapid development of robotics and artificial intelligence, micro-robots will be an efficient tool for the inspection and detection of transformer internal defects. The basic technique, how to accurately and intelligently determine the degree of insulation degradation based on the captured images, is the key to inspection with the micro-robots.

Surface discharge is one of the most common causes of insulation degradation that occurs inside the oil-immersed transformer. It refers to the discharge along the interface of the oil and paper. Figure 1 shows surface discharge near the upper clamp and the tap-connecting lugs on the internal body structure of the transformer. It results in insulation degradation, electric leakage, or even an explosion [5,6]. In recent years, a number of transformer failures caused by surface discharge had serious impacts on power supply and new energy consumption [7,8]. Since surface discharge leads to the carbonization damage of the oil and paper composite insulation, the carbon trace is an important visual characteristic of surface discharge. The area, morphology, and edge features of carbon trace have important reference value for judging and analyzing the cause, degree, and development trend in the surface discharge [9]. Therefore, the accurate semantic segmentation of carbon traces is the premise and foundation for the micro-robot to successfully detect the discharge degree and insulation degradation trend. However, there is no relevant research at present.

As one of the research hotspots in computer vision, semantic segmentation aims to recognize and understand the specific meaning of each pixel in an image. Traditional semantic segmentation is mainly based on low-level features such as texture, color, and shape and then segments the image by clustering or graph cutting, etc. Typical networks include Efficient Graph-Based Image Segmentation, TextonBoost, etc. [10,11]. With the rapid development of deep learning technology, significant breakthroughs have been made in the field of semantic segmentation, and there are some novel and efficient segmentation networks, such as Fully Convolutional Network (FCN) [12], UNet [13], DeepLab [14], etc. To meet the needs of medical image segmentation, UNet was designed to perform pixel accurate localization and segmentation by feature fusion with its special encoding-decoding structure and jump connections [13]. In order to overcome the limitations of UNet’s ordinary convolutional module and achieve a better perception of global features and long-range semantic information, Cao et al. proposed a U-shaped encoder-decoder architecture based on a transformer mechanism. By using a Shifted-Window module to extract contextual features, a Swin-Transformer decoder was designed for the accurate segmentation of the heart and other organ images [15]. Based on the Swin-Transformer, Atek et al. constructed the Transformer Interactive Fusion (TIF) module to realize the fusion of different-scale features and built the dual-scale coding U-type segmentation network SwinT-Unet [16]. Although the transformer module improves the network performance, it also increases the network parameters and decreases the training and inference speed. By using regular convolution in the shallow stage and Tok-MLP module to label and project the convolutional features in the deep stage, UNeXt effectively reduced the network parameters and complexity while achieving a better segmentation performance [17]. Meanwhile, the attention mechanism has provided new insights for segmentation performance improvement. A variety of UNet structures with different attention mechanisms, such as Nested UNet [18], Resnet Coordinate Hardswish UNet (RCH-UNet) [19], and Spatial-Coordinate Attention UNet (SPCA-UNet) [20], have emerged. The above networks perform well in segmenting edge-regular targets such as medical images and traffic road images but face severe challenges in segmenting carbon traces. We put forward a much higher requirement for the segmentation model to perceive the global information and local detail features.

▪Challenges of carbon trace segmentation:

① Inside the metal-enclosed shell of the transformer, the micro-robot needs supplemental light to properly acquire image data. Changes in the intensity of supplemental light lead to large differences in the overall brightness and contrast of the captured images, as shown in Figure 2a,b.

② Changes in the degree of surface discharge result in significant differences in the size of carbon traces, as shown in Figure 2c,d.

③ For the surface discharge, there is spatial randomness of the arc ablation site and local complexity of the dendritic development of carbon traces, resulting in extremely complex edge features, as shown in Figure 2e,f. This trait is the main challenge of carbon trace segmentation, which dramatically reduces the segmentation accuracy.

**Figure 2 sensors-24-06498-f002:**
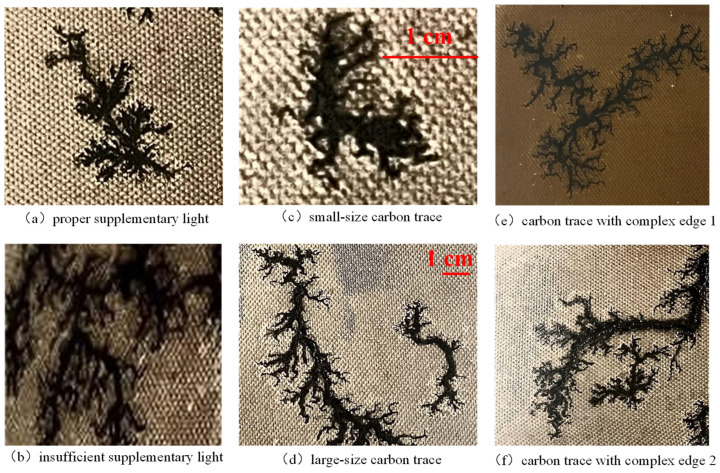
Significant contrast, size differences, and complex edges of the samples.

For the accurate segmentation of carbon traces, an HSP-UNet semantic segmentation network based on the UNet architecture was proposed in this paper. The proposed HSP-UNet model could achieve good semantic segmentation effect for carbon traces with complex edge features and reduce the model parameters and computational overhead.

Main contributions of this paper:

① The grouped HPA module is designed for the high-dimensional feature extraction, which reduces the algorithm complexity while realizing the effective fusion of carbon trace feature maps with different scales.

② To alleviate the semantic gap between encoder and decoder and improve the segmentation completeness of carbon trace, the SCA mechanism is designed to replace the original jump connection.

③ To improve the parsing ability of carbon trace edge features, the PixelShuffle up-sampling module with better adaptability for feature maps is used to replace the original Bilinear Interpolation module.

## 2. Brief Introduction of Our Inspection Micro-Robot

For efficient and convenient inspection of the transformer internal defects, an inspection micro-robot was developed, as shown in Figure 3. The micro-robot mainly consists of a body shell, an ultrasonic emission module, an ultrasonic range module, an image acquisition module, propeller propulsion modules, and a manipulation platform. The body shell of the micro-robot is an elliptical sealed structure used to mount and protect the following functional modules. The ultrasonic emission module is installed at the top of the body, which is mainly used for the three-dimensional positioning of the robot. The image acquisition module is installed on the upper part of the body, which is used to inspect the internal structure of the transformer and collect carbon trace images at the same time. Ultrasonic range modules are installed around the body shell, which are used to detect the distance between the robot and nearby objects. Propeller propulsion modules are used to control the movement of the micro-robot inside the transformer. Meanwhile, there is a manipulation platform for the micro-robot, which can remotely control the robot and store the collected images of carbon traces. The dimensions of the micro-robot are 15 × 15 × 26 cm (Length × Width × Height), which is determined for high throughput of the micro-robot in the narrow space of the transformer.

As the micro-robot inspects the internal structure of the transformer, the image acquisition module will continuously capture the internal environment, and the captured images will be transmitted to the manipulation platform.

## 3. Carbon Trace Image Dataset

### 3.1. Acquisition of Carbon Trace Images

The transformer enclosure is a common site of surface discharge. Due to the difficulty of obtaining carbon trace images of surface discharge inside the actual operating transformer, there are not enough samples. Therefore, an oil–paper insulation discharge test platform was constructed to restore the transformer internal scene and artificially generated carbon trace samples. The test platform mainly consists of a specimen model, a boosting platform, and an image acquisition module. The specimen model consists of a nylon screw, acrylic board, nylon bracket, front electrode, voltage equalization ring, connecting rod, and oil-immersed cardboard, as shown in Figure 4. The size of the oil-immersed cardboard is 25 cm × 15 cm, which is attached to the acrylic plate with a nylon clamp. The tilt angle of the oil-immersed cardboard could be changed by adjusting the angle of the clamp. The boosting platform adopts a transformer (SB-10KVA/100KV, Haotai Technology, Yangzhou, China) to provide the discharge voltage, and the specimen container consists of a transparent acrylic sheet for easy observation and the collection of carbon trace images. The transformer oil used in the test is Keramay # 25 oil. An industrial camera (HTSUA134GC/M, Huateng Vision, Shenzhen, China, 1.3 megapixels, frame rate 211FPS) was used to collect carbon trace images at 25 cm from the oil-immersed cardboard.

Generally, surface discharge will produce two kinds of carbon traces, i.e., dendritic carbon trace and clustered carbon trace. When the oil–paper insulation stays dry, dendritic carbon trace will appear with surface discharge. When the oil-paper insulation is damp, clustered carbon trace appears. In this paper, a total of 499 images of dendritic carbon trace and 565 images of clustered carbon trace were collected. As shown in Figure 5, dendritic carbon trace has a very complex edge, which is the main challenge for accurate semantic segmentation. In contrast, the edge of clustered carbon trace is much smoother, which is relatively much easier to segment.

### 3.2. Image Enhancement Based on the AHE Algorithm

Restricted by the metal-enclosed shell of the oil-immersed transformer, the acquisition of carbon trace often suffers from the problem of insufficient complementary light, resulting in the carbon trace images showing an over-concentration of pixel values, weak contrast, and other problems. At the same time, the oil stains on the oil-immersed cardboard also tend to cause local reflections, which reduces the clarity of carbon trace images. In order to improve the quality of carbon trace images and reduce the difficulty of extracting carbon trace features by the semantic segmentation model, the AHE algorithm was adopted in this paper [21]. Using the distribution function of the cumulative probability of image gray level as the transformation function, the AHE algorithm focuses on the local region of the carbon trace and performs a pixel-by-pixel localized histogram equalization. It can alleviate the over-concentration of pixel values and effectively enhance the contrast of carbon trace images. Due to insufficient supplemental light, the original carbon trace image was dark and low contrast, as shown in Figure 6a. The corresponding gray level probability distribution of the original image was too concentrated, as shown in Figure 6b. After processing with the AHE algorithm, the overall brightness of the image was significantly improved, and the edges of the carbon trace were clearer, as shown in Figure 6c. The processed image showed a uniform distribution of gray levels, indicating that the quality of carbon trace images could be greatly improved with the AHE algorithm, as shown in Figure 6d.

### 3.3. Construction of Carbon Trace Dataset

The acquisition of carbon trace samples inside the oil-immersed transformer is difficult and costly, resulting in the inadequacy of carbon trace samples. In order to improve the segmentation performance and generalization ability of the proposed semantic segmentation model, Gaussian fuzzy process, horizontal and vertical flip, image scale, and horizontal and vertical translation were used to augment the original carbon trace samples. Then, a dataset of the dendritic carbon trace *Set_dentritic_* was constructed, which contained 2495 samples, and a dataset of the clustered carbon trace *Set_cluster_* was constructed, which contained 2825 samples. Furthermore, these two datasets were divided into a training set, validation set, and test set in the ratio of 8:1:1. For the dataset of dendritic carbon traces, the sample sizes in the training set, validation set, and test set were 1996, 250, and 249, respectively. For the dataset of clustered carbon traces, the sample sizes in the training set, validation set, and test set were 2260, 283, and 282, respectively.

## 4. Proposed Network

### 4.1. Network Structure of HSP-UNet

In order to improve the perception of complex edge features and realize the accurate segmentation of carbon trace image, a high-precision semantic segmentation model HSP-UNet was designed based on the structure of a UNet network, introducing the grouped HPA module, SCA attention mechanism, and PixelShuffle up-sampling module. The proposed network consisted of an encoder and a decoder, which were composed of 6-layer down-sampling and 6-layer up-sampling modules, respectively, as shown in Figure 7. The specific design process was as follows:

① Design the grouped HPA module to replace the conventional Conv2d module in the Stage 4~6 layers, which can reduce the number of model parameters and complexity while effectively integrating carbon trace features from different perspectives.

② Design the SCA mechanism to replace the original jump connection, which can alleviate the semantic gap between the encoder and decoder and improve the perception ability for complex edge features of carbon traces, improving the completeness and accuracy of carbon trace segmentation.

③ Use the PixelShuffle module to replace the Bilinear Interpolation (BI) up-sampling module of the decoder, which can help parse the deep semantic features of carbon traces, improving the segmentation accuracy of the complex edge of carbon traces.

**Figure 7 sensors-24-06498-f007:**
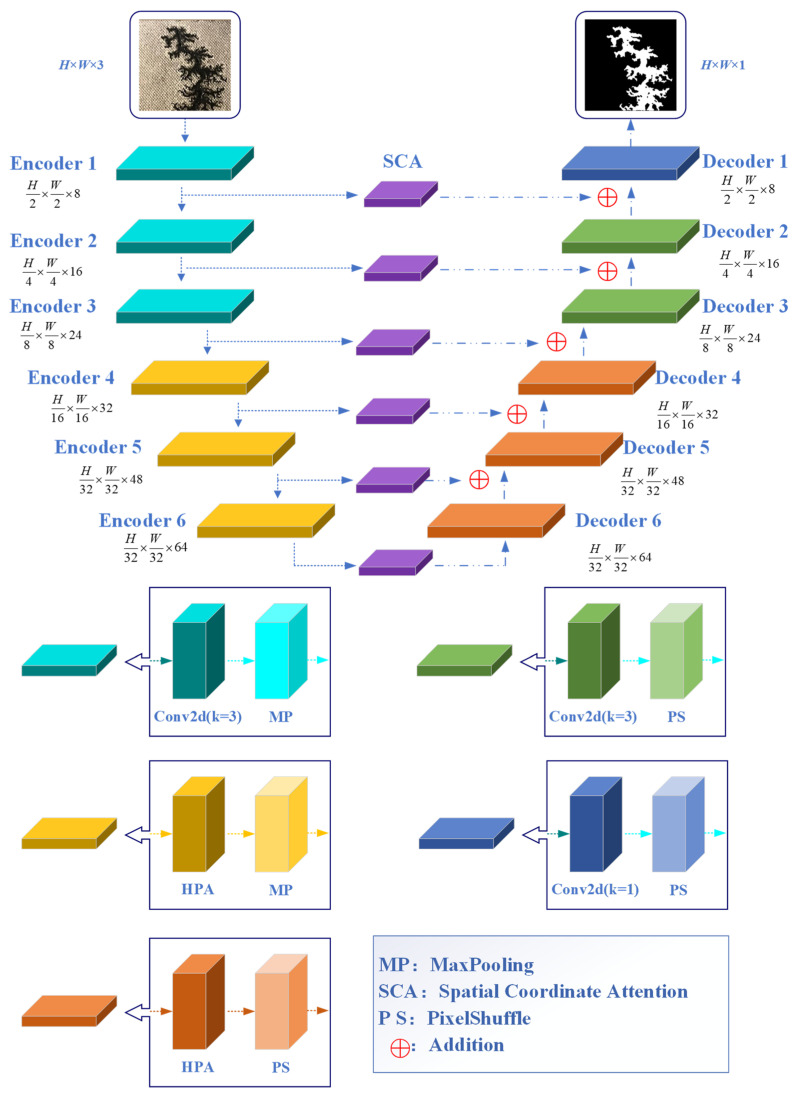
Network structure of the proposed HSP-UNet.

### 4.2. Grouped HPA Module

In order to reduce the model parameters and improve the perception of multi-view features, the HPA module with linear complexity was adopted in this paper. According to the size parameters [B, C, H, W] of the input feature map, a tensor *p* is randomly initialized and adjusted by using the BI algorithm. In order to extract the multi-view features of the input feature map, a grouped HPA module was constructed, which was inspired by the multi-head self-attention (MHSA) mechanism [22], as shown in Figure 8. The input feature map was divided into four groups {X1, X2, X3, X4} along the Channel dimension. The HPA operations were performed on the H-W, C-H, and C-W axes for the first three groups {X1, X2, X3}, respectively. And the depthwise separable convolution (DW) is performed on the fourth group X4. Then, the four groups of feature maps along the Channel dimension were concatenated using the Concat instruction. The merged feature maps were processed by using LN and DW instructions to obtain the final output feature map.

### 4.3. SCA Attention Mechanism

Compared with other attention modules such as SENet [23] and CBAM [24], Coordinate Attention [25] (CA) could better accurately localize and identify the target of interest in the global range, which reduces the loss of spatial positional information as well as the module parameters. The structure of the CA is shown in Figure 9. The input feature maps are averagely pooled in the height direction and width direction, respectively, to obtain the feature maps in two directions, as shown in Equation (1). Then, to obtain the feature maps of 1 × (W + H) × C/r, the processed feature maps are concatenated and sequentially processed with Conv2d, BatchNorm, and Sigmoid instructions, as shown in Equation (2). Next, the feature map is split into two tensors *f^h^* and *f^w^* along the Channel direction and processed with Conv2d and Sigmoid instructions. As a result, the attention weights *g^h^* and *g^w^* on the height and width direction were obtained, as shown in Equation (3). Finally, the *g^h^* and *g^w^* were weighted with the input feature map to obtain the output feature map, as shown in Equation (4) [25].
(1)$$\left\{\begin{array}{l} {Z^{h}}_{c}(h)=\frac{1}{W}\sum_{0\leq i<W}|x_{c}(h,i)\\ {Z^{w}}_{c}(w)=\frac{1}{H}\sum_{0\leq i<H}|x_{c}(j,w) \end{array}\right.$$
(2)$$f=\delta (F_{1}(\left[z^{h},z^{w}\right]))$$
(3)$$\left\{\begin{array}{l} g^{h}=\sigma (F_{h}(f^{h}))\\ g^{w}=\sigma (F_{w}(f^{w})) \end{array}\right.$$
(4)$$y_{c}(i,j)=x_{c}(i,j)\times {g^{h}}_{c}(i)\times {g^{w}}_{c}(j)$$

The discharge carbon trace has obvious spatial edge features, presenting complex edges and rich local information, while its color, texture, and other features are not obvious. So, it is necessary to improve the ability of the CA to perceive the spatial location information. Therefore, the CA-based SCA is proposed to further improve the ability to extract the detailed edge features of carbon traces. First, the input feature layer is globally pooled with the average pool operation and maximum pool operation in the channel direction, respectively. So, the feature layers of the maximum value and the average value at the spatial level are obtained, with the shape of [1, H, W]. Then, the two feature layers of the maximum and the average value are concatenated and fed into the Conv2d layer with the channel number of one, achieving the fusion of spatial location information of the carbon trace. After the operation of activation function, the spatial feature parameters of the input feature layer are obtained. Finally, the above spatial features are weighted with the original input feature layer to enhance the spatial feature of the input layer, which is then sent to the CA mechanism. The structure of the proposed SCA is shown in Figure 10.

### 4.4. PixelShuffle Upsampling Module

Compared with the original BI module in the UNet, the PixelShuffle module is able to learn and optimize its own up-sampling parameters independently, which has better adaptability to the feature map and a better pixel reconstruction effect [26]. For the feature maps of dendritic carbon traces, the PixelShuffle module could better retain the detailed features and boundary information, which helps to improve the semantic segmentation accuracy. In the PixelShuffle module, a *L*-layer convolutional network is used to process the low-resolution feature maps of carbon trace, whose *L* − 1 layers are shown in Equation (5). For the *L*th layer, a convolution with a step size of $\frac{1}{r}$ is used to up-sample the feature maps of carbon trace from the low-resolution space to the high-resolution space, as shown in Equations (6) and (7). And the loss function of the above up-sampling module is the pixel-wise MSE, as shown in Equation (8) [26].
(5)$$\left\{\begin{array}{l} f^{1}(I^{LR};W_{1},b_{1})=\phi (W_{1}*I^{LR}+b_{1})\\ f^{l}(I^{LR};W_{1:l},b_{1:l})=\phi (W_{l}*f^{l-1}(I^{LR})+b_{l}) \end{array}\right.$$
where *W_l_*, *b_l_*, and *l* ∈ (1, *L*−1) are the learnable network weights and bias parameters, respectively; *W*_l_ is a 2D convolution tensor of size *N_L_*_−1_ × *N*_1_ × *K_l_* × *K_l_*, where *N_l_* is the number of features in the *l*th layer, the value of *N*_0_ is C, and *K_l_* is the size of the filter in the *l*th layer; and the bias parameter *b_l_* is a vector with the length of *N_l_*.
(6)$$I^{SR}=f^{L}(I^{LR})=PS(W_{L}*f^{L-1}(I^{LR})+b_{L})$$
(7)$$PS{\left(T\right)}_{x,y,c}=T_{\left\lfloor x/r\right\rfloor ,\left\lfloor y/r\right\rfloor ,C\cdot r\cdot mod(y,r)+C\cdot mod(x,r)+c}$$
where *PS* is a concatenating operator for the periodic pixels that can transform a tensor with the size of $H\times W\times C\cdot r^{2}$ into a tensor with the size of rH×rW×C, as shown in Equation (7); and *W_L_* is a convolution operator with the size of $n_{L-1}\times r^{2}C\times k_{L}\times k_{L}$.
(8)$$l(W_{1:L},b_{1:L})=\frac{1}{r^{2}HW}\sum_{x=1}^{rH}\sum_{x=1}^{rW}{\left(I_{x,y}^{HR}-f_{x,y}^{L}(I^{LR})\right)}^{2}$$

## 5. Results and Discussion

### 5.1. Training Setup

The training environment is Windows 11 × 64, and the hardware parameters are as follows: CPU Intel(R) Core (TM) i5-12500H, RAM 16 GB, GPU Nvidia GeForce RTX2050, video memory 4 GB; and the software parameters are as follows: Python 3.8.17, training framework PyTorch 2.0.1, CUDA version 11.8, CUDNN 8.9.3. The model is trained using an AdamW optimizer, with an initial learning rate of 1 × 10^−3^, a learning rate adjustment strategy of CosineAnnealingLR, a weight decay coefficient of 1 × 10^−2^, the epochs of 300, and a batch size of eight.

### 5.2. Evaluation Metrics

In order to verify and evaluate the performance of the proposed HSP-UNet model, three evaluation metrics, Mean Intersection over Union (MIoU), Pixel Accuracy (PA), and Class Pixel Accuracy (CPA) based on the confusion matrix are used in this paper. The calculations of these three metrics are shown in Equations (9)–(11) [19].
(9)$$I_{\mathrm{mIoU}}=\left(\frac{T_{p}}{T_{p}+F_{p}+F_{n}}+\frac{T_{n}}{T_{n}+F_{n}+F_{p}}\right)\times 100\%$$
(10)$$P_{A}=\frac{T_{p}+T_{n}}{T_{p}+T_{n}+F_{p}+F_{n}}\times 100\%$$
(11)$$C_{\mathrm{PA}}=\frac{T_{p}}{T_{p}+F_{p}}\times 100\%$$
where *T*_p_ is the correctly identified real sample of carbon trace, *F*_p_ is the incorrectly identified real sample of carbon trace, *T*_n_ is the correctly identified sample of the background, and *F*_n_ is the incorrectly identified sample of the background.

### 5.3. Validation of the HSP-UNet

In order to verify the effectiveness of the proposed HSP-UNet, the models UNet [13], UNet++ [27], UNeXt [17], MALUNet [28], and EGE-UNet [22] were used to carry out the comparative analysis. The dataset of dendritic carbon trace *Set_dentritic_* and the dataset of clustered carbon trace *Set_cluster_* were used to train the above models, respectively. As shown in Table 1, among the six segmentation models involved in the comparison, the model parameters and computational GFLOPs of the HSP-UNet were only 0.061 M and 0.066, respectively, which shows a better lightweighting advantage than the other models. The model complexity and arithmetic power demand of the HSP-UNet were comparatively low, which is conducive to the practical deployment of the HSP-UNet. The segmentation effects of all the models on the clustered carbon trace samples were better than that on the dendritic carbon trace samples. The main reason is that the edges of the clustered carbon traces are smoother and easy to segment, while the edges of the dendritic carbon traces are much more complex, which substantially increases the segmentation difficulty. Specifically, HSP-UNet demonstrated obvious performance advantages on both datasets of carbon traces. For the dataset of dendritic carbon traces, HSP-UNet improved the MIoU, PA, and CPA of the benchmark model UNet by 2.13, 1.24, and 4.68 percentage points, respectively. Compared with the other four segmentation models, it also showed a better segmentation performance. For the dataset of clustered carbon traces, the MIoU, PA, and CPA of the six segmentation models were all higher than 90%, 97%, and 94%, respectively. Since the benchmark UNet already had a good segmentation effect on the clustered carbon trace, the MIoU, PA, and CPA of the HSP-UNet are improved by 0.98, 0.65, and 0.83 percentage points, respectively. The enhancement of segmentation performance with the dataset of clustered carbon traces was smaller than that with the dataset of dendritic carbon traces.

The segmentation effects of the dendritic carbon traces and the clustered carbon traces were comparatively analyzed in Figure 11 and Figure 12, respectively. For the dendritic carbon traces, the four models, UNet, UNet++, UNeXt, and MALUNet, were weak in perceiving the edge features and failed to accurately segment the local complex edge, resulting in low values of the MIoU. EGE-UNet adopted the GAB module to replace the jumping connection between the encoder and the decoder, which increased the segmentation accuracy for the edges of carbon trace. However, compared with the GroundTruth, EGE-UNet had poor perception accuracy of local features and lost a large amount of carbon trace details. Compared with the above five models, the HSP-UNet proposed in this paper achieved the best segmentation effect. The SCA mechanism effectively improved the perception of carbon trace edge features, contributing to a better segmentation completeness of carbon trace. And the PixelShuffle upsampling module helped to resolve the carbon trace details, resulting in the refined segmentation of carbon traces, and retained enough detailed information. Due to the smooth edges of clustered carbon trace, the segmentation effects of the six models were less different. However, when focusing on the segmentation effect of the local area with larger curvature (the red circle in Figure 12), the HSP-UNet model still showed a better segmentation performance.

### 5.4. Generalization Performance of the HSP-UNet

To validate the segmentation performance of the HSP-UNet with different light conditions, carbon trace samples with sufficient and insufficient supplementary light were selected from the dendritic and clustered trace datasets. As shown in Figure 13a,b, the HSP-UNet segmented two dendritic carbon traces completely and accurately, with the MIoU of 0.736 and 0.775 for the light-sufficient one and the light-insufficient one. The reason for the relatively lower MIoU value of the light-sufficient sample is that the edge features of this sample are too complex to segment the detail boundaries. With respect to the clustered carbon traces of sufficient and insufficient light, the HSP-UNet also showed a steady segmentation performance, with the MIoU of 0.922 and 0.907, respectively. Similarly, to validate the segmentation performance of the HSP-UNet with samples of different sizes, four carbon traces were selected from the dendritic and clustered datasets. The carbon traces in Figure 14a,c are much larger than that in Figure 14b,d. The segmentation results indicated a good generalization performance of the proposed HSP-UNet with the dendritic and clustered samples, with the MIoU values of 0.774, 0.741, 0.918, and 0.913 from subplot (a) to (d) in Figure 14.

### 5.5. Comparison of Different Attention Mechanism

After verifying the segmentation performance of the HSP-UNet, four attention mechanisms, SCA, SENet [23], CBAM [24], and ECA [29] were selected to compare the segmentation performance for carbon traces. Using the HP-UNet with an addition of the grouped HPA module and the PixelShuffle up-sampling module as the backbone network, the above four attention mechanisms were, respectively, added to analyze the segmentation effect, as shown in Table 2. For dendritic carbon traces with rich detailed features, all four attention mechanisms could improve the segmentation effect, indicating the feasibility of using attention mechanisms to improve the perception of detailed features. Among them, the SCA mechanism achieved the best improvement, with the MIoU, PA, and CPA improved by 2.19, 1.34, and 4.78 percentage points, respectively. For clustered carbon traces, the improvements of the four attention mechanisms were relatively small. The reason was that for the clustered carbon traces with smoother edges and fewer detailed features, the attention mechanisms could not give full play to their detail perception capability. Considering the segmentation needs of the dendritic and clustered carbon traces, the SCA mechanism was adopted in this paper.

### 5.6. Ablation Tests for the HSP-UNet

The model validation and the comparative analysis of four attention mechanisms showed that the HSP-UNet proposed in this paper had the best segmentation performance of carbon traces. Taking the dendritic carbon trace with a higher segmentation difficulty as the object, the ablation test was carried out to analyze the contributions of the grouped HPA module, SCA, and the PixelShuffle. The ablation test would provide a reference for the subsequent model improvement and design. Using the UNet model as the benchmark, the test results were shown in Table 3, where √ indicated that the corresponding module was used. Compared with the benchmark UNet, the adoption of the grouped HPA module improved the MIoU, PA, and CPA of carbon trace segmentation by 0.61, 0.12, and 1.02 percentage points, respectively; the addition of the SCA could improve the MIoU, PA, and CPA by 0.79, 0.11, and 1.97 percentage points, respectively; and the adoption of the PixelShuffle module improved the MIoU, PA, and CPA by 0.20, 0.01, and 0.69 percentage points, respectively. Therefore, the SCA mechanism contributed the most to the performance improvement, the grouped HPA module the second, and the PixelShuffle module the least.

Meanwhile, the Grad-CAM was used to visually compare and analyze the contribution of different modules to the segmentation effect of carbon traces. The multi-dimensional feature map output from the last convolutional layer was utilized to generate the heat maps of the ablation test. With respect to the original image of Sample 1 in Figure 15a, the benchmark UNet had poor ability to perceive carbon traces in Figure 15b. The activation of the upper right carbon trace was low, resulting in an incomplete segmentation of the carbon trace region. The addition of the HPA module improved the segmentation completeness of the carbon trace, but the activation of the carbon trace region was still low, as shown in Figure 15c. The SCA mechanism dramatically improved the segmentation completeness and activation intensity of the carbon trace. However, the activation of the background region in the left part of the image was high, which may induce the model to misclassify the background as the carbon trace, as in Figure 15d. The PixelShuffle module had a lower activation intensity than the SCA mechanism, but it also reduced the activation of the background region, which was conducive to reducing the misclassification probability of the background, as in Figure 15e. The combined use of the above three modules could effectively improve the activation intensity of the carbon trace while reducing the activation intensity of the background region, which is conducive to improving the segmentation integrity and accuracy of carbon traces, as in Figure 15f. 

Similar to Sample 1, there were obvious differences in the Grad-CAMs with different modules for Sample 2. For the benchmark UNet, the activation intensity of the lower part of the carbon trace was low, resulting in an incomplete segmentation of the carbon trace, as in Figure 15h. As shown in Figure 15i, the HPA module improved the perception completeness of the carbon trace, but the activation intensity of carbon trace edges was still low. This can lead to significant loss of detailed edge information and make the network misclassify the pixels near the carbon trace edges. In Figure 15j, the SCA module significantly improved the activation intensity and completeness of the carbon trace, but the activation intensity of the background near the carbon trace edge was a little high. It indicated that the perception of the carbon trace edge for the network with the SCA module needs to be further improved. As shown in Figure 15k, the PixelShuffle module can be a good complementary for the SCA module, because the activation intensity of the background near the carbon trace edges was lower than that with the SCA module, obtaining a much clearer boundary. Meanwhile, the overall activation intensity of the carbon trace with the PixelShuffle module was lower than that with the SCA module. Finally, by combining the three modules mentioned above into the benchmark UNet, the Grad-CAM obtained a good activation intensity and completeness with a clear boundary between the carbon trace and the background, as shown in Figure 15l. The above Grad-CAM results were in good agreement with the evaluation indexes in Table 3.

### 5.7. Discussion

Through the forementioned analysis, the proposed HSP-UNet outperformed over five State-of-the-Art segmentation models. But the segmentation performance on the dendritic carbon traces needs to be further improved. In subsequent studies, the following optimizations may be worth carried out: (1) The conventional convolution kernel in the grouped HPA module has a fixed rectangular inception field, which shows an insufficient adaptation to multi-scale complex edge features of the dendritic carbon traces. Owing to the deformable inception field, deformable convolution [30] may have a better feature extraction ability. (2) The U-shaped architecture is difficult to balance shallow spatial features and deep semantic features. Spatially detailed features are usually sacrificed to ensure the overall accuracy requirements of semantic segmentation, resulting in the need to improve the segmentation performance of the U-shaped model on the dendritic carbon traces. New model architectures, such as Bisenet series [31], may be an effective way to improve the segmentation performance with carbon traces.

## 6. Conclusions

Aiming at the accurate assessment of surface discharge inside the transformer, this paper constructed the HSP-UNet semantic segmentation network by means of an AHE-based image enhancement and network structure design and optimization based on the UNet, which achieved a good semantic segmentation of carbon traces with complex edge features. It would provide technical support for an accurate assessment of the transformer insulation condition.

(1)Aiming at the over-concentration of pixel values and the weak contrast of carbon trace images collected inside the transformer, the AHE algorithm was used for image enhancement, which effectively reduced the extraction difficulty of carbon trace features. At the same time, four data augmentation methods were used to construct the dataset of dendritic carbon trace containing 2495 samples and the dataset of clustered carbon trace containing 2825 samples.(2)With the goal of model lightweighting and accurate segmentation, the HSP-UNet model was constructed by integrating the grouped HPA module, SCA mechanism, and PixelShuffle module. Experimental results showed that the model parameter and GFLOPs were only 0.061 M and 0.066, respectively, which showed a good lightweighting advantage. Meanwhile, compared with the existing models, HSP-UNet had better segmentation on both carbon trace datasets. For dendritic carbon traces, HSP-UNet improved the MIoU, PA, and CPA of the benchmark UNet by 2.13, 1.24, and 4.68 percentage points, respectively. For clustered carbon traces, HSP-UNet improved the MIoU, PA, and CPA by 0.98, 0.65, and 0.83 percentage points, respectively. Similarly, the validation experiments with the samples of different light conditions and different size demonstrated a good generalization performance of the proposed HSP-UNet.(3)Ablation experiments for dendritic carbon traces showed that the grouped HPA module, the SCA mechanism, and the PixelShuffle module adopted in the proposed HSP-UNet can all improve the segmentation effect. Due to the improvement in the ability to perceive detailed features, the SCA mechanism contributed the most to the model performance, improving the MIoU, PA, and CPA by 0.79, 0.11, and 1.97 percentage points, respectively.

## Figures and Tables

**Figure 1 sensors-24-06498-f001:**
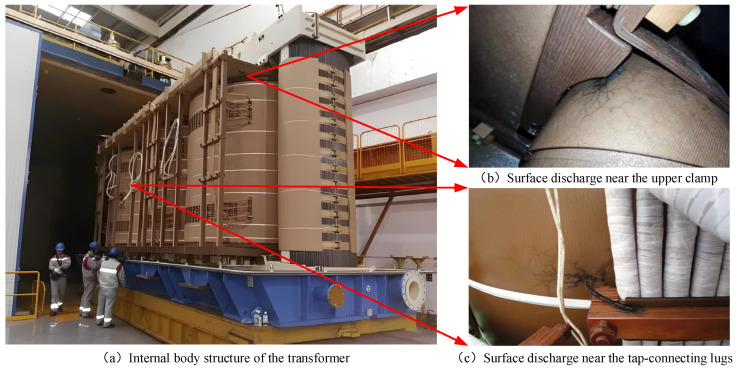
Surface discharge and carbon traces of different parts inside the transformer.

**Figure 3 sensors-24-06498-f003:**
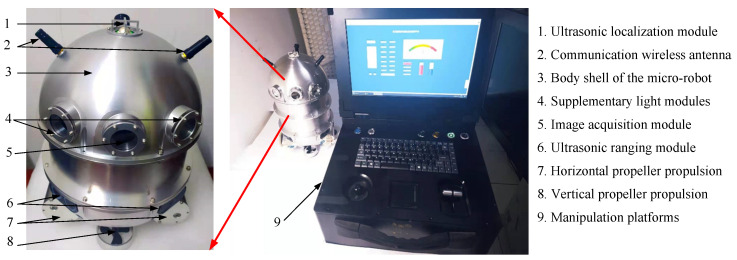
The micro-robot for transformer internal inspection.

**Figure 4 sensors-24-06498-f004:**
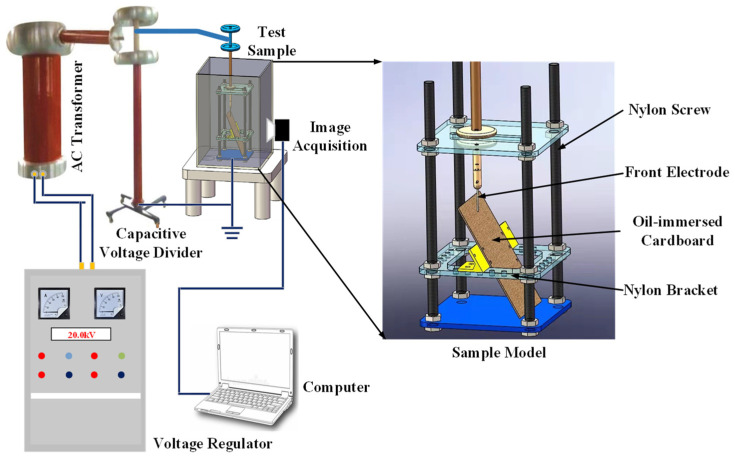
Test platform for carbon trace image acquisition.

**Figure 5 sensors-24-06498-f005:**
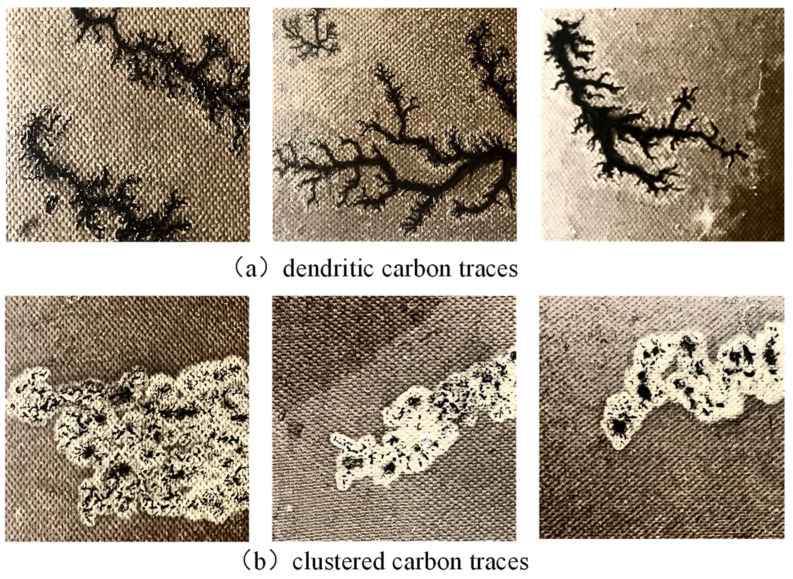
Examples of two kinds of discharge carbon traces.

**Figure 6 sensors-24-06498-f006:**
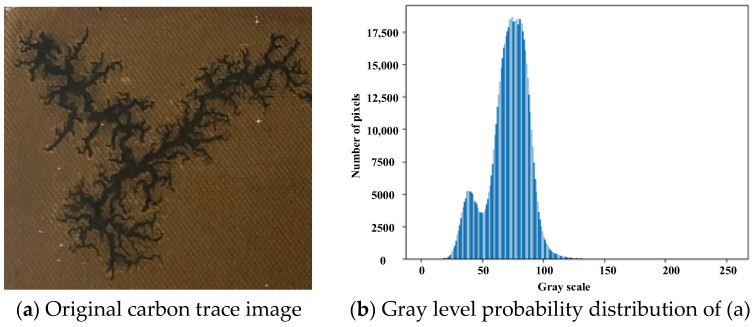
Comparison of carbon trace image with and without the AHE.

**Figure 8 sensors-24-06498-f008:**
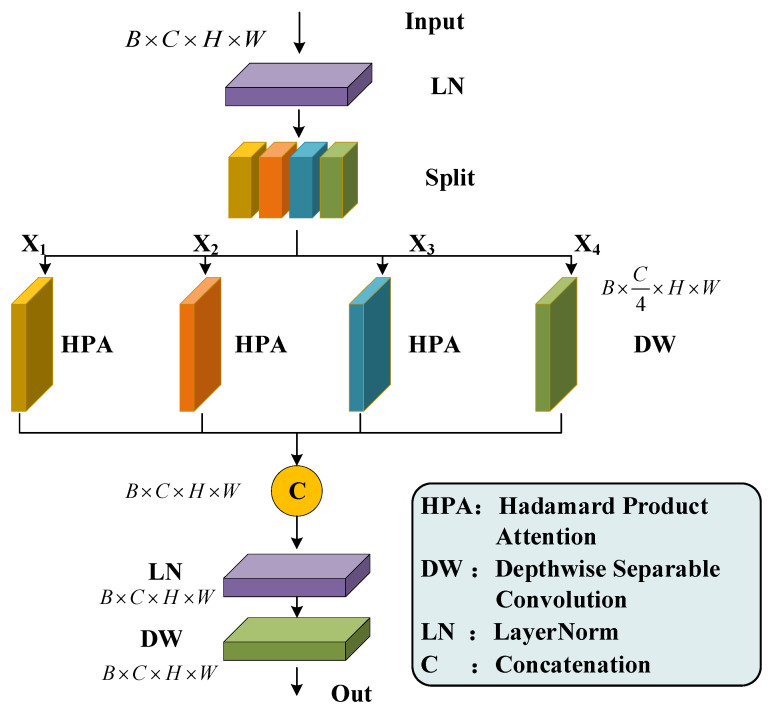
Structure of the grouped HPA module.

**Figure 9 sensors-24-06498-f009:**
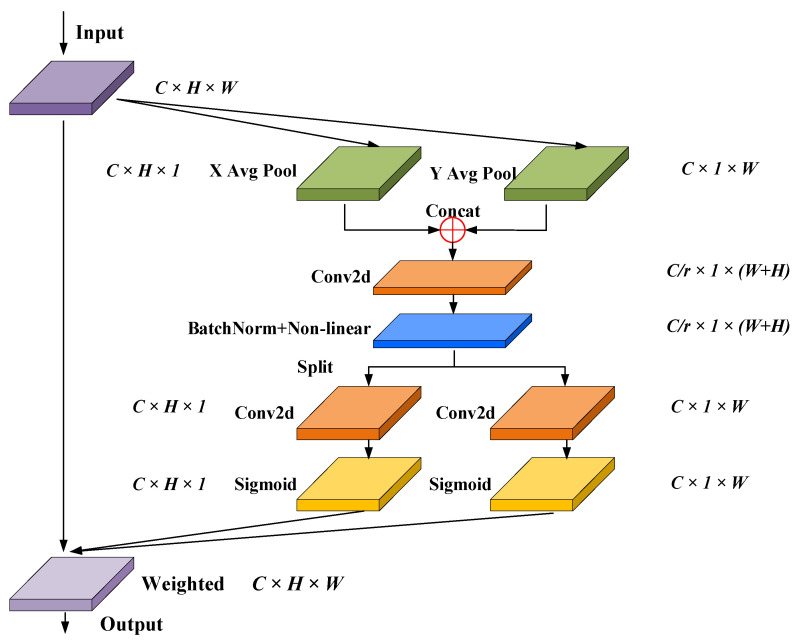
Structure of the CA module.

**Figure 10 sensors-24-06498-f010:**
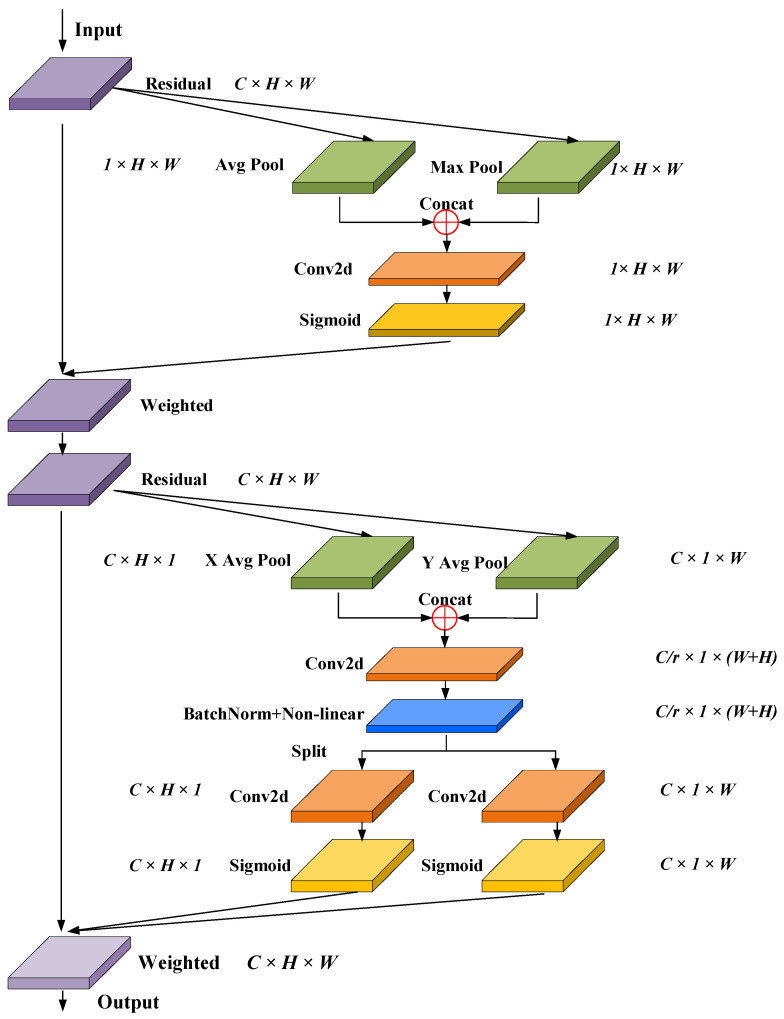
Structure of the SCA.

**Figure 11 sensors-24-06498-f011:**
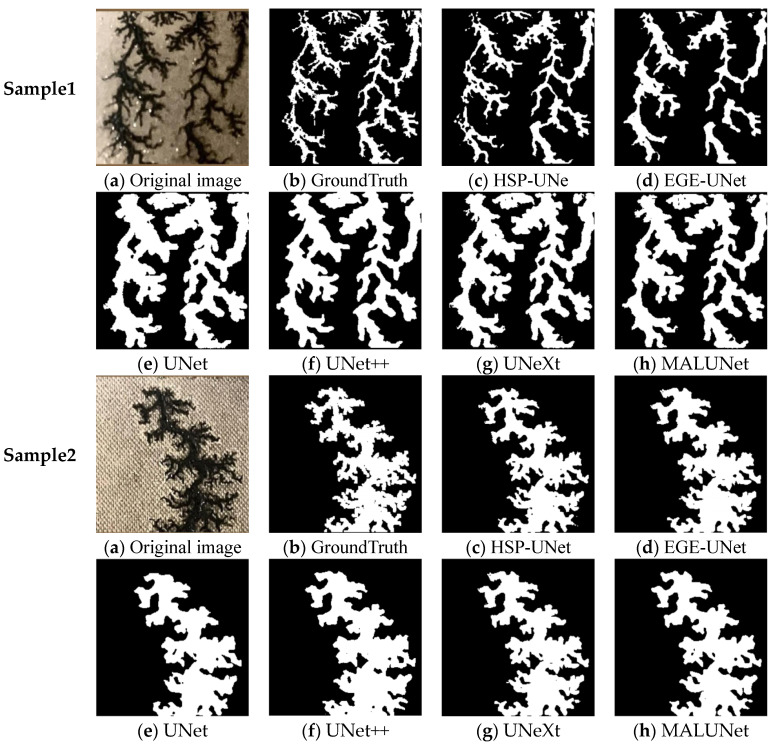
Segmentation comparison of the dendritic carbon traces.

**Figure 12 sensors-24-06498-f012:**
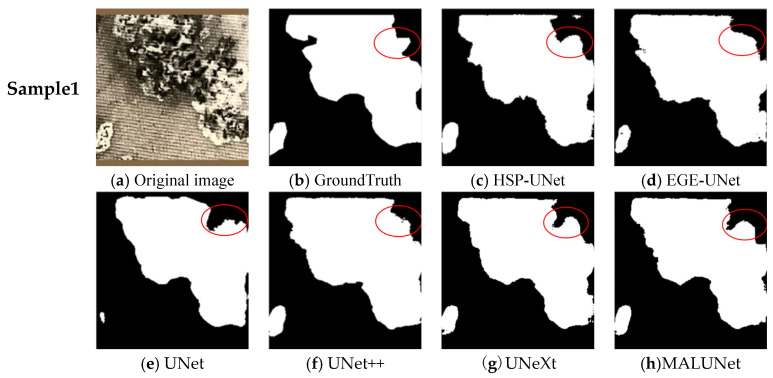
Segmentation comparison of the clustered carbon traces.

**Figure 13 sensors-24-06498-f013:**
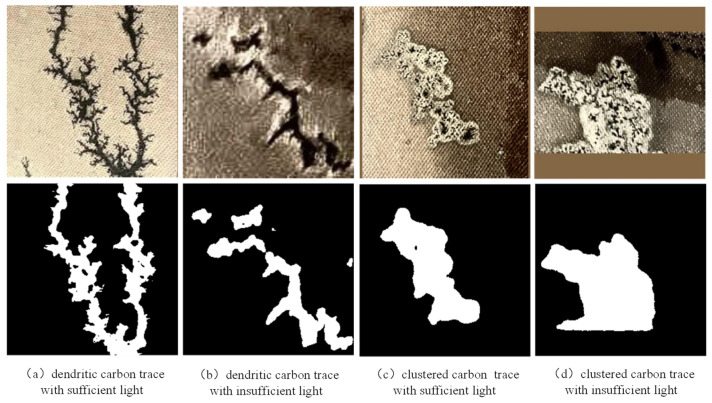
Segmentation performance with samples in different light conditions.

**Figure 14 sensors-24-06498-f014:**
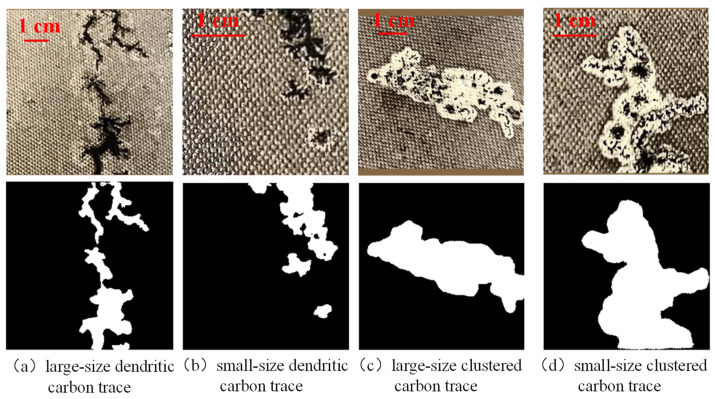
Segmentation performance with samples of different sizes.

**Figure 15 sensors-24-06498-f015:**
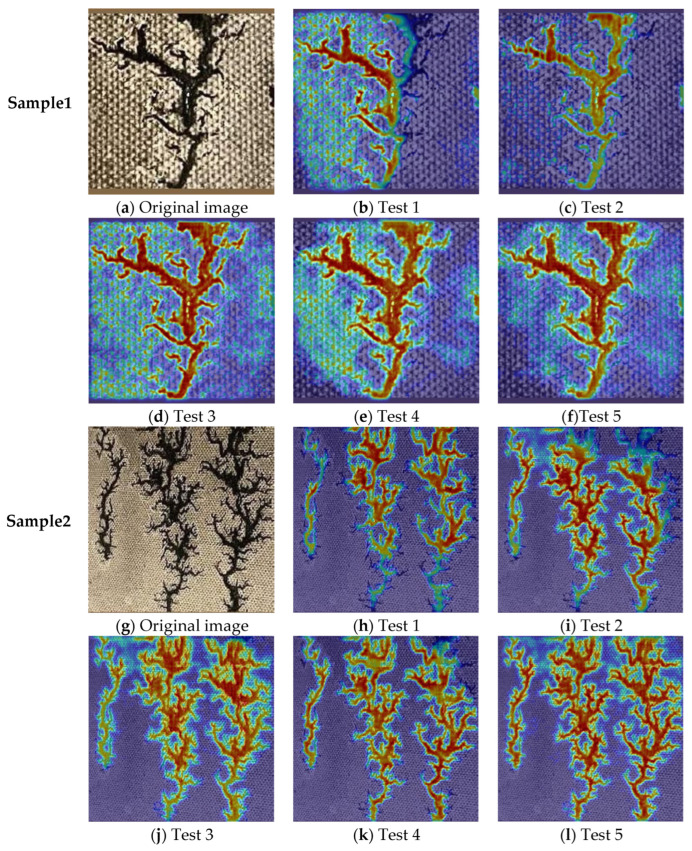
Grad-CAM comparison of the HSP-UNet ablation test.

**Table 1 sensors-24-06498-t001:** Segmentation comparison of carbon traces with 6 models.

Dataset	Model	Params↓	GFLOPs↓	*I*_mIoU_ (%)	*P*_A_ (%)	*C*_PA_ (%)
*Set_dendritic_*	UNet (Base)	31.2 M	13.76	73.53	93.17	84.42
	UNet++	9.2 M	34.86	73.47	93.07	84.32
	UNeXt	1.5 M	0.57	73.93	93.58	85.47
	MALUNet	0.177 M	0.085	74.15	93.79	86.23
	EGE-UNet	0.053 M	0.072	74.71	94.10	86.30
	**HSP-UNet**	**0.061 M**	**0.066**	**75.66**	**94.41**	**89.10**
*Set_cluster_*	UNet (Base)	31.2 M	13.76	90.41	97.42	94.56
	UNet++	9.2 M	34.86	90.43	97.53	94.77
	UNeXt	1.5 M	0.57	90.54	97.44	94.53
	MALUNet	0.177 M	0.085	91.14	97.58	95.13
	EGE-UNet	0.053 M	0.072	91.21	98.01	95.24
	**HSP-UNet**	**0.061 M**	**0.066**	**91.39**	**98.07**	**95.39**

**Table 2 sensors-24-06498-t002:** Segmentation comparison of four attention mechanisms.

Types	*Set_dentritic_*			*Set_cluster_*		
	*I*_mIoU_ (%)	*P*_A_ (%)	*P*_E_ (%)	*I*_mIoU_ (%)	*P*_A_ (%)	*P*_E_ (%)
HP-UNet	74.37	94.07	86.32	90.55	97.73	94.97
HP-UNet+SEnet	74.54	94.34	85.06	90.74	97.84	94.73
HP-UNet+CBAM	75.02	94.30	87.70	91.25	**98.12**	95.24
HP-UNet+ECA	75.34	94.41	87.72	91.35	98.01	**95.41**
HP-UNet+SCA	**75.66**	**94.41**	**89.10**	**91.39**	98.07	95.39

**Table 3 sensors-24-06498-t003:** Ablation results of the HSP-UNet.

Num	UNet	HPA	SCA	PixelShuffle	*I*_mIou_ (%)	*P*_A_ (%)	*P*_E_ (%)
1	√				73.53	93.17	84.42
2	√	√			74.47	94.29	86.44
3	√	√	√		75.46	94.40	88.41
4	√	√		√	75.37	94.31	87.92
5	√	√	√	√	75.66	94.41	89.10

## Data Availability

The data presented in this study are available on request from the corresponding author.

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
