# Peer review of "HSP-UNet: An Accuracy and Efficient Segmentation Method for Carbon Traces of Surface Discharge in the Oil-Immersed Transformer"

_sensors, 2024, doi:10.3390/s24196498_

Round 1

Reviewer 1 Report

Comments and Suggestions for Authors

1. The purpose of image enhancement is to make the samples more diverse, not more uniform. Does the AHE algorithm help improve generalization of the model?

2. How is the Ground Truth of segmentation obtained?

3. It is difficult to say whether innovative points proposed in this article are really effective. On the one hand, the test dataset is too small, only a few hundred, and on the other hand, the experimental results are not significant, as shown in Table1 and Table 2. It is likely that only a few sample swings caused changes  in the indicators.

Comments on the Quality of English Language

English expression needs some minor modifications

Reviewer 2 Report

Comments and Suggestions for Authors

Manuscript: HSP-UNet: An Accuracy and Efficient Segmentation Method for Carbon
Traces of Surface Discharge in the Oil-Immersed Transformer

Journal: Sensors

The topic contains significant scientific value, both on the developed sensor and on the methods used in processing the images obtained in the cases of dendritic and clustered carbon traces. While the authors have prepared a commendable manuscript, several modifications are needed to ensure its suitability for publishing in "Journal of Sensors."

The below notices outline the necessary revisions for the improved version:

1.     Adding more keywords can make the study more apparent, such as "surface discharge,", “image segmentation”.

2.     All equations used should be based on references.

3.     The side images in Figure 1 are unclear and not well explained either in the title of the figure or in the text of the manuscript.

4.     All abbreviations should be known before using them. The authors have used many abbreviations such as FCN (Fully Convolutional Network) and many others.

5.     This text should be cited to an appropriate reference “To meet the needs of medical image segmentation, UNet was designed to perform pixel accurate localization and segmentation by feature fusion with its special encoding-decoding structure and jump connections. ”

6.     The authors mentioned several advantages of Combined (HSP-UNet), but they also need to refer to drawbacks and limitations.

7.     The authors frequently use terms such as 'we' and 'our,' which are generally not preferred in academic writing. It is recommended to adopt a more formal, objective tone.

8.     It is preferable to add a schematic drawing that shows the parts of the micro-robot and the connections between its parts.

9.     The micro-robot dimensions must be included.

10.  Although there is a slight difference between the number of images obtained for dendritic and clustered carbon traces, why is there this difference in the number of images?

11.  It is important to clarify what the vertical and horizontal axes represent in the algorithm diagrams in Figure 6.

12.  It is not clear what the authors mean by the reference to the presence of 2,495 dendritic carbon trace samples and 2,825 clustered carbon trace samples and how this relates to the number of images previously reported for each type of carbon trace.

13.  When looking at Table 3, the ImIou(%), PA(%), and PE(%) values did not change significantly in all five cases.

14.  Despite the effective presentation of the results, no relevant references supported the authors' interpretations.

Comments on the Quality of English Language

In general, the English language in the manuscript is simple and understandable, with the exception of the frequent use of abbreviations, which confuses the reader.

Reviewer 3 Report

Comments and Suggestions for Authors

This article continues the series of research conducted by Hongxin Ji, is an interesting work about micro-robot. The manuscript is devoted to the development of innovative architecture Hadamard specially designed for carbon trace segmentation, which is certainly a relevant topic. To address the pixel over-concentration and weak contrast of carbon trace image, the Adaptive Histogram Equalization algorithm (AHE) is used for image enhancement. It should be noted that the presented study is important in view of the fact that large oil transformers play a crucial role in the power system, and due to their complex internal structure, internal defects are quite difficult to detect.

1.       The authors write that the commonly used methods to detect internal defects in large transformers during maintenance are ineffective and not very accurate. I believe that the authors should comment in the manuscript on how cost-effective their proposed method is.

2.       Chapter 2 «Brief Introduction of Our Inspection Micro-Robot» needs to be significantly shortened and references should be made to articles that already have a brief description of the micro-robot, since the author has already published several articles devoted to the micro-robot. For example, the article 10.3390/s24134309

The results here are certainly of sufficient interest to deserve publication in Sensors once the comments have been corrected.

Round 2

Reviewer 1 Report

Comments and Suggestions for Authors

The author responded well to the comments

Comments on the Quality of English Language

English needs to be further improved

Author Response

Comment: English needs to be further improved

Response: Thank you for your suggestion. We have revised the manuscript according to your suggestion. The revised have been highligted in red in the manuscript.

Reviewer 2 Report

Comments and Suggestions for Authors

The revised version of the manuscript "HSP-UNet: An Accuracy and Efficient Segmentation Method for Carbon Traces of Surface Discharge in the Oil-Immersed Transformer" has become better, and the authors' responses to the reviewers' queries were generally good. Although I find the manuscript worthy of attention and publication, I urge the authors to mention the drawbacks combined (HSP-UNet) to make the manuscript more useful.

In the previous review, I asked the question below, but the authors still need to provide an answer (perhaps they forgot to do so).

Comment 6: The authors mentioned several advantages of Combined (HSP-UNet), but they also need to refer to drawbacks and limitations.

However, this does not harm the manuscript's acceptance, although adding it would have been better. 

Author Response

Comment 6:The authors mentioned several advantages of Combined (HSP-UNet), but they also need to refer to drawbacks and limitations.

Response:We deeply apologize for the omission of a response to comment 6.  Thank you for your kindly reminder and suggestion. We have added a new section 5.7 Discussion to adress the limitations of the HSP-UNet and introduce the plan of future studies. The following paragraph is the text of section 5.7: Through the forementioned analysis, the proposed HSP-UNet outperformed over five state-of-the-art segmentation models. But the segmentation performance on the dendritic carbon traces needs to be furtherly improved. In subsequent studies, the following optimizations may be worth carried out: (1) The conventional convolution kernel in the grouped HPA module has a fixed rectangular inception field, which shows insufficient adaptation to multi-scale complex edge features of the dendritic carbon traces. Owing to the deformable inception field, deformable convolution [33] may have better feature extraction ability. (2) The U-shaped architecture is difficult to balance shallow spatial features and deep semantic features. Spatially detailed features are usually sacrificed to ensure the overall accuracy requirements of semantic segmentation, resulting that the segmentation performance of the U-shaped model on the dendritic carbon traces remains to be improved. New model architectures, such as Bisenet series [34], may be an effective way to improve the segmentation performance with carbon traces.